

# Observation of horizontal temperature variations by a spatial heterodyne interferometer using single-sided interferograms

Konstantin Ntokas[1], Jörn Ungermann[1], Martin Kaufmann[1], Tom Neubert[2], and Martin Riese[1]

[1]Institute of Energy and Climate Research (IEK-7: Stratosphere), Forschungszentrum Jülich, Jülich, Germany
[2]Central Institute of Engineering, Electronics and Analytics (ZEA-2: Electronic Systems), Forschungszentrum Jülich, Jülich, Germany

**Correspondence:** Konstantin Ntokas (k.ntokas@fz-juelich.de)

**Abstract.** Analyses of the mesosphere and lower thermosphere suffer from a lack of global measurements. This is problematic because this region has a complex dynamic structure, with gravity waves playing an important role. A limb sounding spatial heterodyne interferometer (SHI) was developed to obtain atmospheric temperature retrieved from the $O_2$ A-band emission, which can be used to derive gravity wave parameters in this region. The 2-D spatial distribution of the atmospheric scene is

captured by a focal plane array. The SHI superimposes the spectral information onto the horizontal axis across the line-of-sight (LOS). In the usual case, the instrument exploits the horizontal axis to obtain spectral information and uses the vertical axis to get spatial information, i.e. temperature observations at the corresponding tangent points. This results in a finely resolved 1-D vertical atmospheric temperature profile. However, this method does not make use of the horizontal across LOS information contained in the data.

In this manuscript a new processing method is investigated, which uses single-sided interferograms to gain horizontal across LOS information of the observed temperature field. Hereby, the interferogram is split and each side is mirrored at the center of the horizontal axis. Each side can then be used to retrieve an individual 1-D temperature profile. The location of the two retrieved temperature profiles is analysed using prescribed horizontal temperature variations, as it is needed for deriving wave parameters. We show that it is feasible to derive two independent temperature profiles, which however will increase the

requirements of an accurate calibration and processing.

## 1 Introduction

The dynamical structure of the mesosphere and lower thermosphere (MLT) is mainly driven by atmospheric waves like planetary waves, tides and gravity waves (Vincent (2015)). Gravity waves are small to medium scale wave patterns, which transport energy from lower altitudes to the MLT region. Commonly known sources of gravity waves in the lower atmosphere are the

uplift of air masses due to orography, convection, and unstable jets. Alexander et al. (2010) summarize the current treatment of gravity waves in global circulation models, where unresolved gravity waves are mostly parameterized. Becker and Vadas (2020) point out that gravity waves can also have sources in higher altitudes which are not well understood, but can have a large effect on the MLT region. Chen et al. (2022) underline the importance of gravity wave observations by multiple examples and outlines an observation method where gravity wave parameters can be extracted from temperature fields.





Temperature in the MLT region can be derived by measuring the $O_2 (0,0)$ atmospheric A-band emissions at 762nm as shown by Ortland et al. (1998) and Sheese (2009). A limb sounding instrument was developed to measure these emissions (Kaufmann et al., 2018). After a successful launch of the first instrument version, a second version has been developed in the recent years. It is partly described by Chen et al. (2022) and will be described further in Sec. 2 and Sec. 3.1. Using this instrument, the temperature can be derived from relative intensities of the $O_2 (0,0)$ atmospheric A-band emission lines. Thus, no absolute

radiometric calibration is needed, which facilitates the calibration of the instrument. The emission is visible during day- and night-time. The instrument is highly miniaturized and energy efficient so that it can fly on nanosatellites, so called CubeSats (Poghosyan and Golkar (2017)). CubeSats allow a cost-efficient way to design and operate satellites due to the utilization of largely standardized components.

     Our instrument works as a camera where the atmospheric scene is mapped onto the detector. A spatial heterodyne interfer-

ometer (SHI) is used to detect the spectrum of the $O_2$ A-band emission and superimposes the spectral information in horizontal direction across the LOS. In the usual processing, the horizontal direction is used to extract spectral information and vertical direction gives spatial information. This allows to extract a finely resolved 1-D temperature profile from one image, which can be subsequently used to derive wave parameters as described by Ern (2004) and applied for our instrument by Chen et al. (2022).

40       To exploit some of the spatial information in horizontal direction, we propose a new processing method which allows to retrieve two 1-D temperature profiles from one image by using single sided interferograms and mirroring them at the center. Johnson et al. (1996) and Gisi et al. (2012) already used mirrored interferograms for the far-infrared spectrometer (FIRS)-2 and for the TCCON FTIR spectrometer, respectively, to achieve a higher resolution. In our case, we use the single-sided interferogram to gain horizontal information of the measured atmospheric scene. Chen et al. (2022) showed that medium-scale

gravity waves can be resolved, if it it possible to obtain two temperature profiles from a single interferogram. The methodology to fulfill this requirement is described in this paper.

     A thorough precision analysis is key to assess the limitations of this method. Therefore we detail the precision budget of the data product, in particular with regard to signal-to-noise limitations at the upper boundary of the measurement domain.

     The paper is structured as follows. We introduce the instrument in Sec. 2. An overview of the simulation setup is given in

Sec. 3, containing the interferogram simulation presented in Sec. 3.1, followed by the detailed forward simulation in Sec. 3.2 and the data processing in Sec. 3.3. The temperature dependency of the emission lines and the resulting temperature precision are discussed in Sec. 4. An extended discussion on using half interferograms is given in Sec. 5. Hereby, we look at the temperature sensitivity of the retrieval with respect to the temperature variation in horizontal direction in Sec. 5.1. Further, we asses the locations of the retrieved temperatures using half interferograms for simulated horizontal temperature variations

in Sec. 5.2. At last we asses the effect of apodization onto the retrieval using half interferograms in Sec. 5.3.





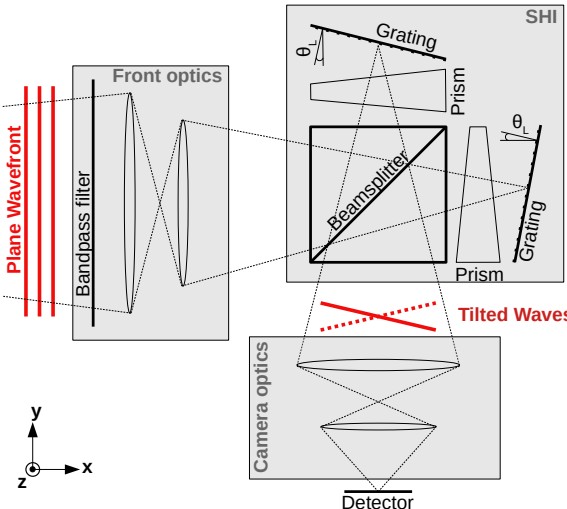

**Figure 1.** Schematic of the SHS instrument

## 2  Spatial heterodyne interferometer

A spatial heterodyne interferometer (SHI) is similar to a Michelson interferometer, but the two mirrors are replaced by fixed tilted gratings. This measurement method was firstly developed by Connes (1958) and with the subsequent availability of imaging detectors, it was further developed to remote sensing methods by e.g. Harlander et al. (1992), Cardon et al. (2003), Watchorn et al. (2001). Figure 1 shows a schematic of the instrument. The incident light is imaged by the front optics onto diffraction gratings, after passing through a beam splitter. The camera optics images the gratings onto a 2-dimensional focal plane array (FPA). The tilt angle of the wave fronts and therefore the frequency of the interference pattern on the FPA is dependent on the frequency of the incoming light. Multiple emission lines result in superimposed cosine waves across the FPA in x-direction. The FPA is two dimensional, so that the spatial distribution of the atmospheric scene is maintained throughout the instrument. One measurement therefore contains spatial information along the z-axis and superimposed spectral and spatial information along the x-axis. Note that the z- and x-axis correspond to the vertical and horizontal across-track dimension in the atmosphere.

## 3  Simulation setup

An interferogram simulation is presented in Sec 3.1. The details of the forward simulation to calculate the expected signal is introduced in Sec. 3.2. At last the data processing is described in Sec. 3.3.





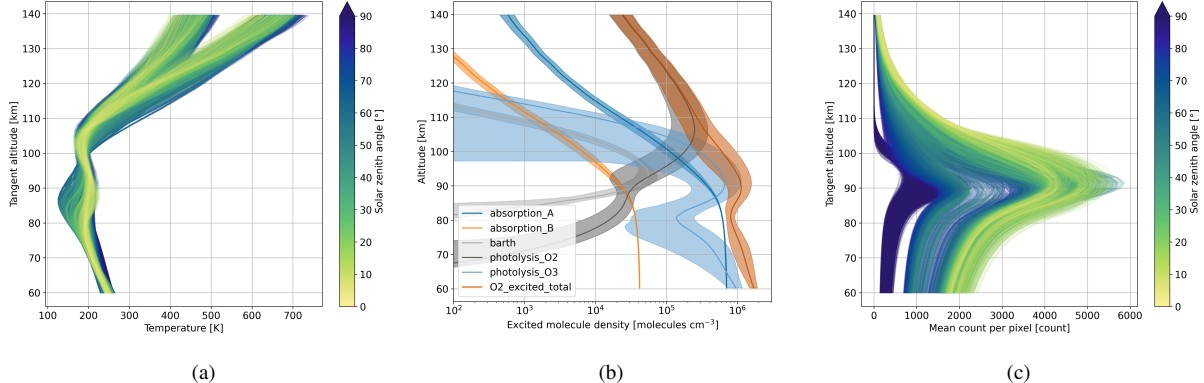

(a)                   (b)                   (c)

**Figure 2.** (a) 1-D input temperature profiles used in the forward calculation; (b) number density of of excited $O_2$ molecules due to the five production mechanisms; total excited $O_2$ is the sum of the five parts; (c) estimated count per pixel received at the detector; Solar zenith angle above $90°$ referring to night-time simulation; the integration time is set to 1s and 10s for day- and night-time simulation, respectively; HAMMONIA data set is used for the forward simulations (Schmidt et al., 2006)

## 3.1 Interferogram simulation

The mathematical derivation for an interferogram measured by an SHI is presented by Harlander et al. (1992), Smith and Harlander (1999), and Cooke et al. (1999). Chen et al. (2022) give more details on the mathematics for the instrument described here.

A line-by-line model is used to simulate spectra which are converted to interferograms. A 1-D interferogram with some horizontal variation is defined by

$$I(x) = \sum_{i=1}^{I} S_i(x)\left[1 + \cos(2\pi f_i x)\right], \tag{1}$$

where $S_i(x)$ is the radiance variation across the horizontal field of view and $f_i$ is the spatial frequency for a given emission line $i$. The spatial frequency corresponds to the wavenumber by $f_i = 4(\sigma_i - \sigma_L)\tan\theta_L M$, where $\sigma_L$ and $\theta_L$ are the Littrow wavenumber and Littrow angle, respectively. The Littrow angle is the tilt of the gratings as shown in Fig. 1. Note, that the Littrow wavenumber corresponds to zero spatial frequency. $M$ is the magnification factor of the camera optics and $\sigma_i$ is the wavenumber of the emission line. This instrument uses a CMOS based detector, where shot noise dominates the precision of the data (Liu et al., 2019). Shot noise can be described by a Poisson process with mean and variance equal to the signal. For the expected signal levels, this can be approximated by an additive white Gaussian noise with standard deviation equal to the square root of the signal in each pixel. The specification of the current instrument version is given in Tab. 1.





**Table 1.** Summary of instrument specification

| Parameter | Property |
| --- | --- |
| Littrow wavenumber | $13047\text{cm}^{-1}$ |
| Littrow angle | $6.6°$ |
| Magnification factor of camera optics | 0.57 |
| Groove density of gratings | $300\text{mm}^{-1}$ |
| Spectral range | $13059\text{cm}^{-1}$ to $13166\text{cm}^{-1}$ |
| Field of view | $1.3\text{deg}^2$ |
| | ($\approx 60\text{km}^2$ for orbit altitude of 600km) |
| Etendue | $0.018\text{cm}^2$ sr |
| Grating efficiency | 0.8 at 762nm |
| Detector | GSENSE400BSI |
| Detector number range | $0 - 4095$ (12bit) |
| Detector columns/rows | 860/860 |
| Pixel pitch | $11\mu\text{m}$ |
| Quantum efficiency | 0.7 at 762nm |

## 3.2 Detailed forward simulation

This section introduces shortly the simulation of the $O_2$ A-band emission, which is described by Chen et al. (2022) in detail. The $O_2$ A-band is a electronic transition from the excited state $O_2(b^1\Sigma_g^+, v = 0)$ to the ground state $O_2(X^3\Sigma_g^-, v = 0)$ centered at 762nm in the near-infrared. The band consists of multiple emission lines due to the transition of multiple rotational states, which absorptions and emissions between the rotational states are calculated using the HITRAN database (Gordon et al. (2022)). The photo-chemical processes producing the excited $O_2$ molecules can be put in three groups. The first group is the absorption in the B-band and the A-band itself. The second group produces excited $O_2$ molecules via collisional energy transfer with highly excited molecules and atoms produced by photolysis of $O_2$ and $O_3$. The third is called Barth-process and is a purely chemical source, which was first described by Barth and Hildebrandt (1961). It is independent of solar radiation and it is therefore the only process active during night-time. A detailed description of the dayglow $O_2$ A-band excitation is given by Sheese (2009), Bucholtz et al. (1986), Zarboo et al. (2018) and Yankovsky and Vorobeva (2020).

To calculate the $O_2$ A-band emission, data of the HAMMONIA (Hamburg Model for the Neutral and Ionized Atmosphere) run of Schmidt et al. (2006) is used. It gives monthly averaged 3-D data sets for temperature, pressure and the volume mixing ratios of the required substitutes involved in the photo-chemical processes for solar minimum and maximum conditions. The overall description of the used one-dimensional radiative transfer model is described by Chen et al. (2022). The model is used to generate test data sets to be used in the following study, covering a equidistant longitude and latitude grid of $10°$. At each longitude and latitude point the solar zenith angle is calculated by assigning the timestamp to noon at the 15th of each month.





This is done for each month and for solar minimum and maximum conditions. The expected mean signal per pixel at the detector can be calculated by

$$105 \quad S_p(z) = \frac{S_r(z)\, E\, L\, t_{int}}{N} \tag{2}$$

where $S_r$ is the spectrally integrated radiation within the spectral range of the instrument for a given tangent altitude $z$, $E$ the etendue of the instrument, $L$ the loss factor or efficiency of the instrument (28% including grating efficiency, quantum efficiency of the detector and loss of 50% due to beam splitter), $t_{int}$ the integration time and $N$ is the total number of pixels of the 2d detector array. The input temperature profiles are shown in Figure 2a. The deviation in two branches above 120km corresponds to the solar minimum and maximum conditions. Based on the temperature profiles, Figure 2b shows the mean and standard deviation of the expected number density of excited $O_2$ molecules due to the different production mechanisms. The total excited $O_2$ correspond to the expected number density of day-time conditions. The Barth-process corresponds to the night-time conditions. Propagating this through the radiative transfer model and applying Eq. (2), where we assume 1s and 10s for day- and night-time, respectively, we can calculate the average count per pixel measured by the detector. The results are shown in Figure 2c. The night-time simulations show a lower variability than the day-time simulations. This can be explained by the larger variation of the photolysis production mechanisms shown in Figure 2b, which is mainly caused by the varying solar zenith angle. Note that during day-time the expected signal at the instrument decreases below 80km even though the number density of excited $O_2$ increases below that. This can be explained by self-absorption, which limits the information content of temperature data obtained from measurements of that region. The decreasing signal in higher altitudes above 120km constraints the measurement method from above. Further discussion on the upper limit is given in Sec. 4. To neglect self-absorption in this paper, we will focus on a field of view between 80km and 140km.

### 3.3 Fast data processing

The first step in the data processing is to subtract the non-modulated part, which corresponds to the low frequencies in the spectrum. Subsequently applying a Fourier transformation along the x-axis, the interferogram is converted into a spectrum. A 2-dimensional radiative transfer model is used to simulate across-track temperature variations. Assuming that self-absorption is small, we can focus on the area close to the tangent layer, where most of the information comes from when integrating along the line of sight. Note that this holds true for tangent altitudes above 80km for the $O_2$ A-band emission.

A simplified forward model then can be used to solve the inverse problem and retrieve temperature. Instead of calculating the full radiative transfer equation, it takes the HITRAN data set to get the relative distribution of the oxygen emission lines for a given temperature, convolves the emissions with a given instrument line shape (ILS) and scales the total spectrum with a scaling factor. We thus retrieve an averaged temperature from the horizontal temperature variation for a given tangent altitude.

The ILS of a finite interferogram is defined by a sinc-function which resolution is determined by the length of the interferogram. To minimize the side lobes of the sinc function, apodization is commonly applied in Fourier spectrometry resulting in a smoother spectral output. It increases the localization of the spectral information, which can also help to be more robust against





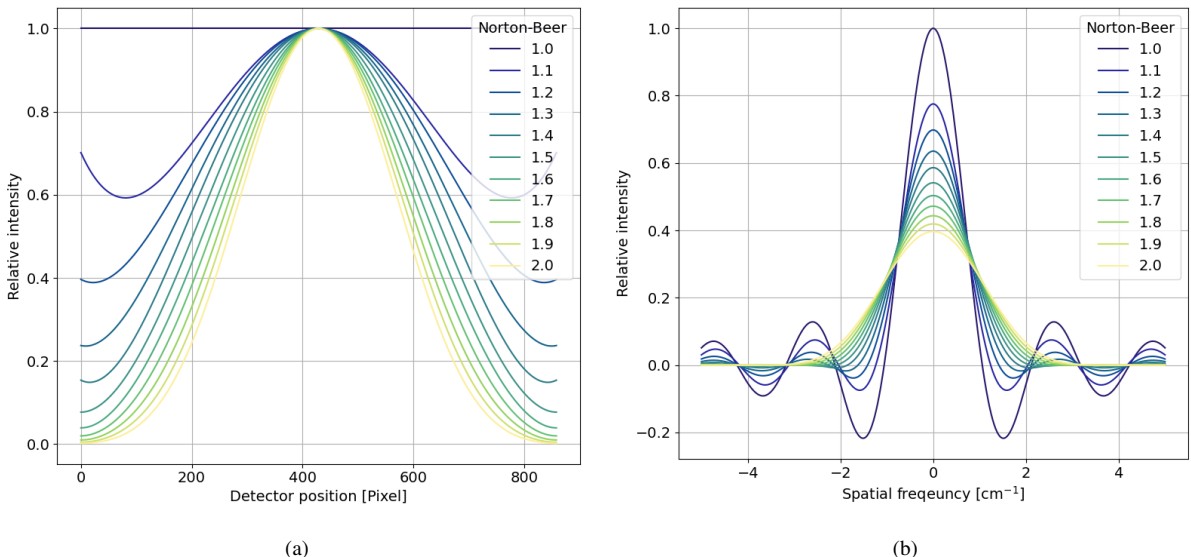

**Figure 3.** Apodization functions used for the assessment; (a) apodization function in the spatial domain; (b) apodization function in the spectral domain;

instrumental errors. However, apodization decreases the spectral resolution and it is therefore a trade off between spectral resolution and decrease of the side lobes. Filler (1964) introduced the Filler's diagram, which plots these two measures against each other. It is often used to assess the performance of given apodization functions. Norton and Beer (1976) and Naylor and Tahic (2007) show that the Norton-Beer apodization has the best properties. The extended version given by Naylor and Tahic (2007) is shown in Figure 3 for the spatial and spectral domain. 1.0 refers to no apodization, and 1.2, 1.4 and 1.6 refers to the weak, medium and strong apodization given by Norton and Beer (1976). Note that the number refers to the full width half maximum (FWHM) of the apodization function relative to the sinc function and thus higher number means stronger apodization.

## 4 Temperature dependency of the $O_2$ A-band emission

Our simplified forward model relies on the temperature dependent rotational distribution of the $O_2$ A-band emission. The temperature dependency is shown in Figure 4a. At low temperatures the central frequencies show higher intensities, which decrease to the sides. Higher temperatures show a more broadened distribution which entails a decrease of the integrated intensity within the bandpass filter. The gradient with respect to temperature decreases with higher temperatures, which entails a decrease in sensitivity towards higher temperatures. Convolving the emission lines with an instrument line shape (ILS) corresponding to a Norton-Beer strong apodization (Sec. 3.3), results in the spectra presented in Figure 4b. Note the increase of the intensities at $13165 \text{cm}^{-1}$ due to the high density of emission lines in the R-branch.





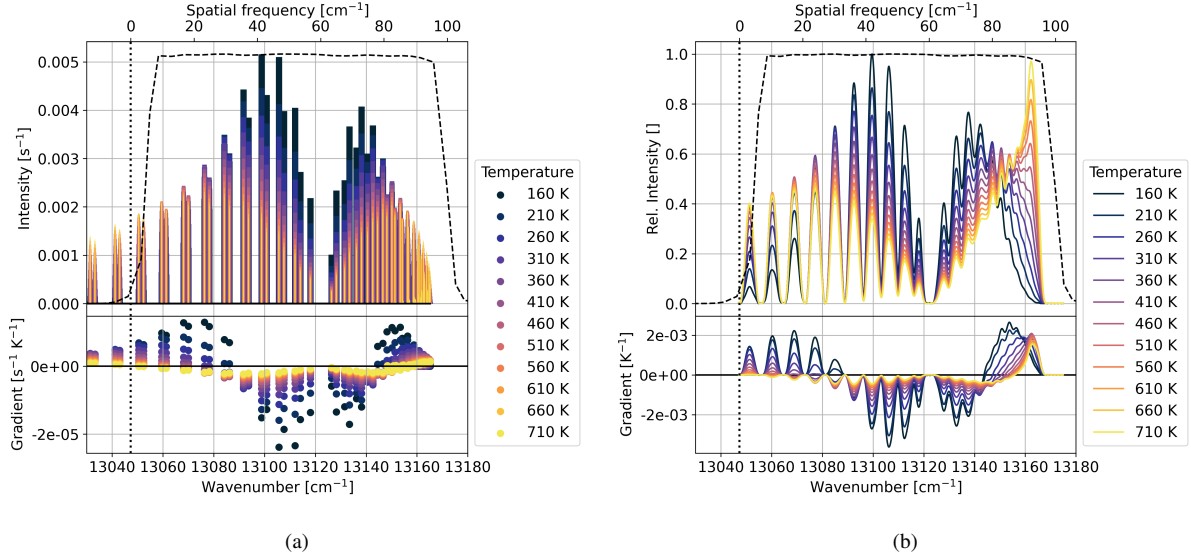

**Figure 4.** Rotational distribution of the $O_2$ A-band emission; the dashed and vertical dotted line indicate the curve of the bandpass filter and the Littrow wavenumber, respectively; the top x-axis shows the spatial frequency at the detector; the gradient is calculated by finite difference along the temperature axis; (a) Intensity of emission lines calculated using HITRAN for different temperatures; (b) convolution of emission lines with Norton-Beer strong ILS;

In the following, we propagate the shot noise in the interferogram through the simplified temperature retrieval for different signal levels and constant temperatures. For each signal level and temperature, we perform a Monte-Carlo simulation with 300 samples, simulate an ideal interferogram using Eq. (1) for a given signal level and temperature, apply shot noise approximated by additive white Gaussian noise with variance equal to the signal level, and retrieve temperature as presented in Sec. 3.3. The signal level is expressed in the signal-to-noise ratio (SNR), which is the ratio of the mean to the standard deviation thus the

square root of the signal level itself. Figure 5 shows the bias and the standard deviation of the retrieved temperatures obtained in the Monte-Carlo simulations. The top axis shows the mean signal assuming that 20 rows are accumulated, which increases the SNR by a factor of $\sqrt{20}$. The main reason for the binning is the limited data downlink capacity of micro-satellites. A binning of 20 rows results in a vertical spatial sampling of approximately 1.5km. The lower sensitivity of the the $O_2$ A-band emission with higher temperatures is also here visible. For higher temperature either a higher SNR is needed to keep the precision at the

same level, or one needs to accept a lower precision if SNR stays constant. The bias for low signals in Figure 5a is caused by using magnitudinal spectra. When taking the the absolute value of the spectrum, the spectral noise follows a Rice distribution (Talukdar and Lawing, 1991) which deviates from the normal distribution for spectral values close to zero. Therefore, low signals are affected more. More details on the spectral noise of a magnitudinal spectrum is given in Appx. A.

    We then evaluate the interpolated temperature precision field presented in Figure 5b on the expected signal levels (Figure 2c) for typical temperature profiles in the MLT region (Figure 2a). Hereby, we assume a binning of 20 rows. The results are shown





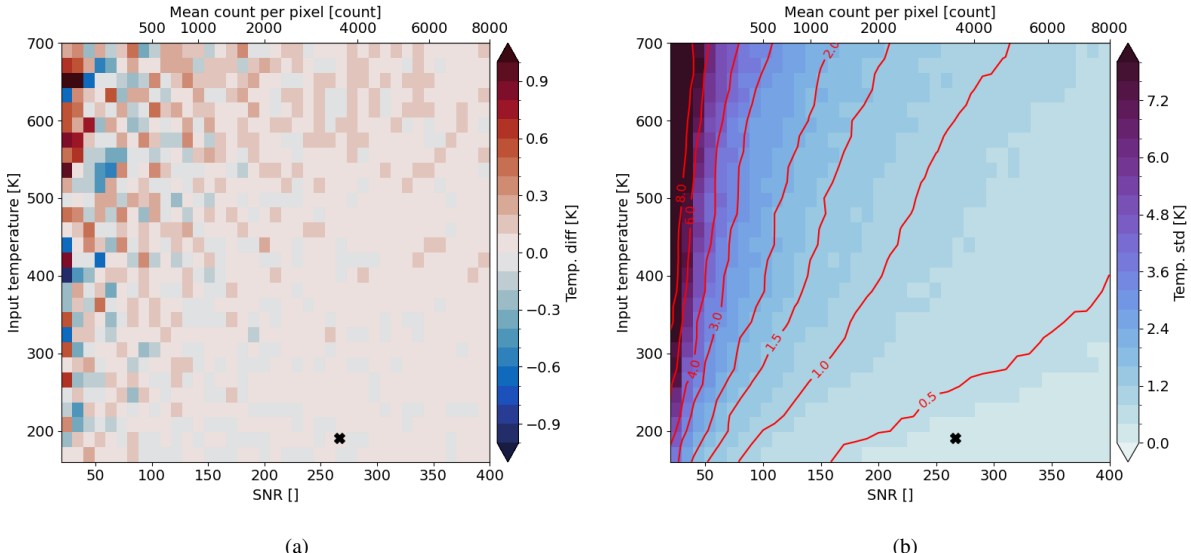

|(a)| |(b)|

**Figure 5.** Results of the Monte-Carlo simulations with 300 samples where shot noise is propagated into the temperature retrieval for different SNRs and temperatures; (a) mean temperature differences; (b) standard deviation of the retrieved temperatures; contour lines are calculated on smoothed data; the marker indicate the mean expected SNR and temperature in the mesopause region at 90km altitude

in Figure 6. The night-time simulation gives an temperature precision below 1K for tangent altitudes between 85km and 100km. For the day-time simulation, a temperature precision of around 0.5K can be achieved in the strong signal layer between 85km and 95km. The temperature precision then decreases for higher tangent altitudes due to the decrease in signal and higher temperatures.

In the next step, we asses the required number of binning rows to achieve a certain temperature precision. Hereby, we extract a contour line from the temperature precision field in Figure 5b, get the required signal for the temperatures presented in Figure 2a and calculate the required binning using the expected signal in Figure 2c. The results for different temperature precision levels are shown in Figure 7. During day-time we expect to resolve the higher altitudes on average up to approximately 105km, 115km, 125km and 135km with a temperature precision of 1K, 2K, 4K and 8K, respectively, when a binning of 20 175    rows is applied. During night-time simulation, accepting low temperature precision does not help to resolve higher altitudes because the signal decreases strongly above 100km, as shown in Figure 2c. In general, a larger binning can be applied to get more accurate results at the cost of spatial resolution, which is already proposed by Florczak et al. (2022). It is therefore a trade-off between spatial sampling and temperature precision.





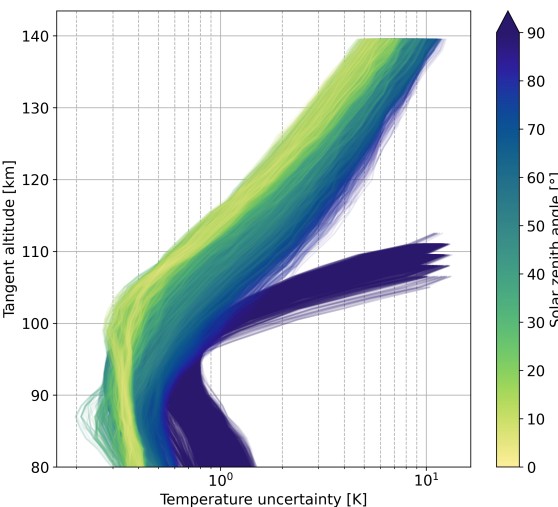

**Figure 6.** Temperature precision calculated from the expected signal from Figure 2c and the temperatures from 2a and using an interpolated temperature precision field presented in Figure 5b assuming a binning of 20 rows; note that signal counts below 100 counts are not considered which cuts off the night-time simulations around 110km

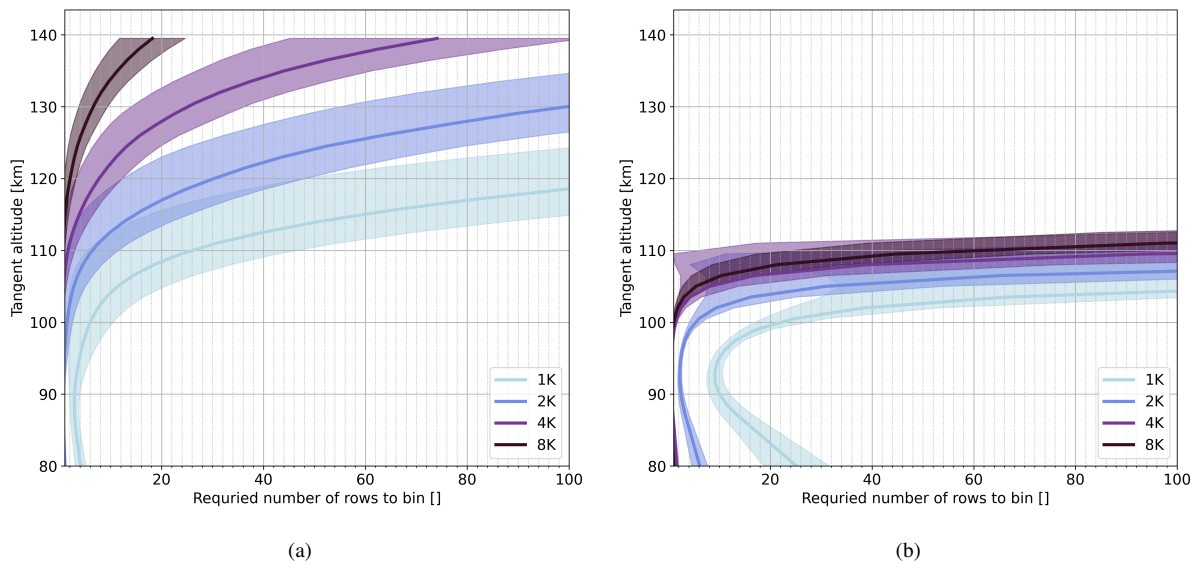

**Figure 7.** Required binning of rows to achieve a certain temperature precision (a) during day-time and (b) during night-time; the solid line shows the mean and the shaded area the standard deviation of all used samples from Figure 2c



## 5 Obtaining horizontal spatial information by interferogram splitting

As explained in Sec. 2, the instrument contains the 2-D spatial distribution of temperature in its field of view. To gain more information, one can split the interferogram at the zero optical path difference (ZOPD). Exploiting the symmetry of the interferogram, each side is then mirrored around the ZOPD. This results in an interferogram of equal length as using the full interferogram, which then entails the same spectral resolution. When using the full interferogram, one half of the shot noise propagates into the real and the other half into the imaginary part of the spectrum. Mirroring the interferogram however causes

the shot noise to be symmetric, so that it is then fully propagated into the real part of the spectrum, resulting in a higher noise level by a factor of $\sqrt{2}$. To show this numerically, we perform a Monte-Carlo simulation with 300 samples, where we simulate an interferogram using Eq. (1) for a constant temperature of 200K across the horizontal field of view. The signal level is set to a SNR of 100, which results in a temperature precision of approximately 1K as seen in Figure 5b. Figure 8a shows the mean and the standard deviation of the interferogram samples. In the following step, we run through the processing chain explained

in Sec. 3.3, extract the spectral noise and retrieve the temperature for each Monte-Carlo sample. The results are shown in Figure 8b. As a reference we show the spectrum without noise. Further, we show the standard deviation of the noise of the 300 samples for the case using the full interferogram and the two cases using mirrored single-sided interferograms. As stated before, the noise level is higher by a factor of $\sqrt{2}$ for the mirrored single-sided interferograms compared to the full interferogram. This results in a correspondingly increased temperature precision of 1.4K. Note that normally the noise is evenly distributed

in the spectrum. If the signal however is close to zero, the noisy signal can take positive and negative values. Considering only magnitudinal spectra, the negative values are mirrored, resulting in a smaller standard deviation. More discussion on this is given in Appx. A.

Performing the same analysis for multiple SNR and temperature levels gives the temperature precision of the right single-sided interferograms depicted in Figure 9a. Figure 9b shows that for most temperature and SNR levels it holds true that the

temperature precision is decreased by a factor of $\sqrt{2}$ when using only the right single-sided interferogram. The same simulation has been performed for the left single-sided interferogram and showed a similar result. Thus, we can conclude that the increase of spectral noise by a factor of $\sqrt{2}$ results in a decreases temperature precision by the same factor.

To study the influence of horizontal temperature variation on the temperature retrieval, we look at a simple example first. A linear temperature gradient of 20K over the horizontal field of view of 60km, shown in Figure 10a, is incorporated into the

interferogram by using Eq. (1). Figure 4a shows that for higher temperatures the integrated intensity within the bandpass filter decreases as the distribution of the emission lines becomes flatter. Following Eq. (1), the interferogram is just the sum of cosine waves with amplitude and offset equal to the intensity of the emission lines. The interferogram's baseline therefore also shows a decrease with higher temperatures, as shown in Figure 10b. It shows an interferogram without noise and without temperature gradient as reference. We also show the difference between the interferogram with the linear temperature gradient and the

reference interferogram relative to the mean signal of the reference interferogram. The linear regression of the difference agrees well with the relative variation of the integrated intensity within the bandpass filter. We perform a Monte-Carlo simulation with 300 samples with shot noise modelled in the interferogram space. We run through the processing chain as explained





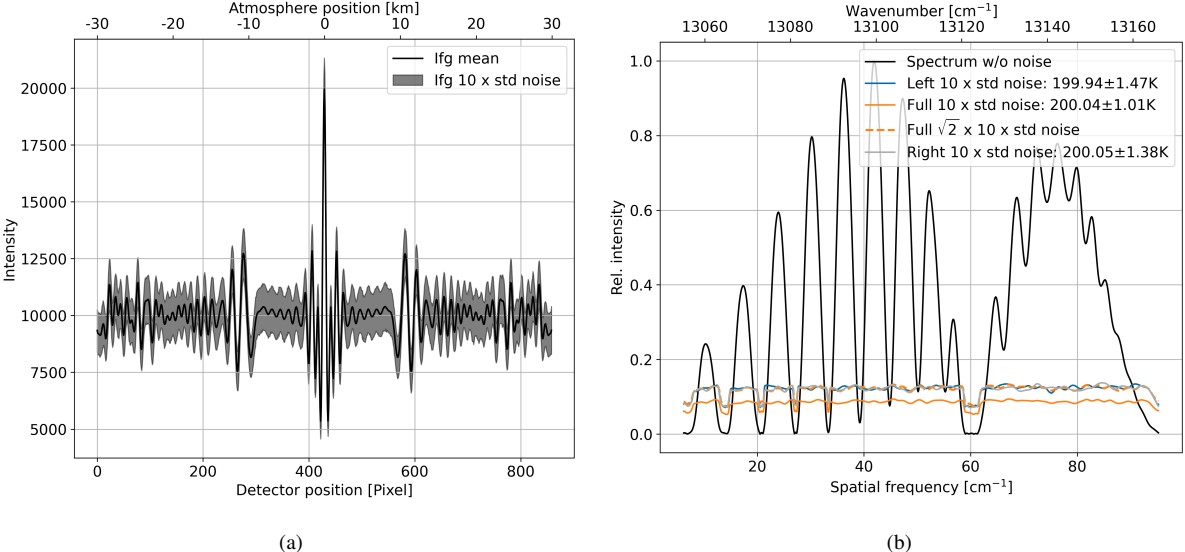

(a)                                                                          (b)

**Figure 8.** Results of Monte-Carlo simulations with 300 samples where shot noise is propagated through the spectrum into the temperature retrieval for an interferogram with a constant temperature of 200K; (a) mean and $10\times$ the standard deviation of the 300 interferogram samples; (b) spectrum without noise as a reference; $10\times$ the standard deviation of the spectral noise using full interferograms and left- and right-side of the interferograms; standard deviation of the noise using the full interferograms multiplied by a factor of $\sqrt{2}$ is shown as dashed line; Norton-Beer strong apodization is applied;

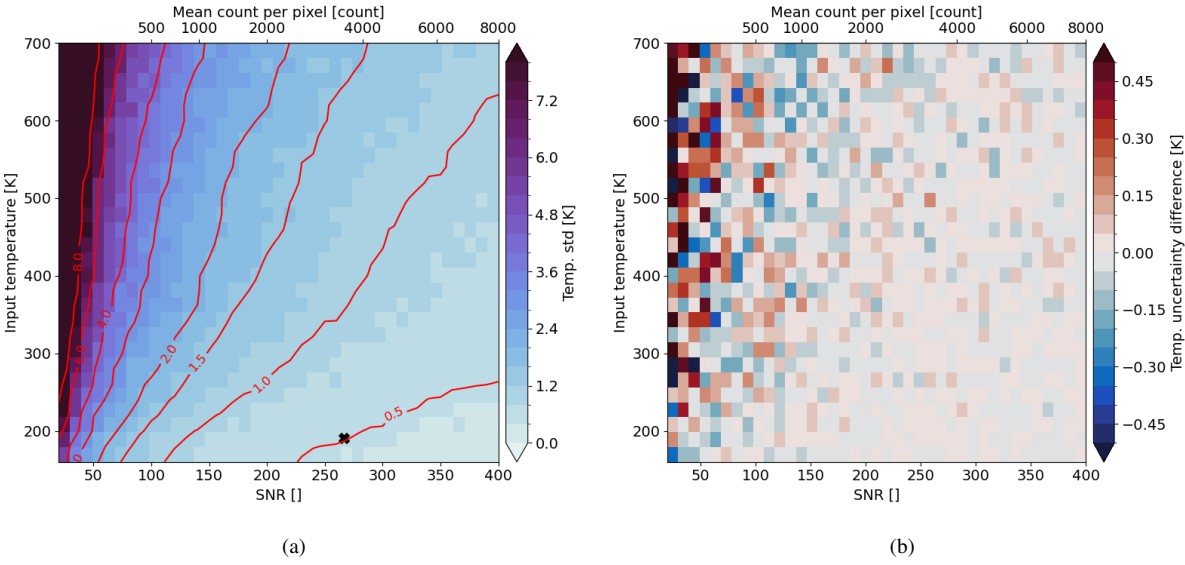

(a)                                                                          (b)

**Figure 9.** (a) Temperature precision of using the right single-sided interferograms assessed for different SNRs and temperatures; (b) difference of temperature precision using the right single-sided interferograms and temperature precision of full interferogram multiplied by $\sqrt{2}$;





in Sec. 3.3 to get a spectrum and a retrieved temperature for each sample. Note that the temperature gradient results in a tilted non-modulated part which is fitted and subtracted before splitting the interferogram. This is performed for the case
using the full interferogram as well as the left and right mirrored interferogram, separately. Figure 10c shows the mean of all noisy spectra with constant temperature gradient as a reference. Furthermore, the smoothed mean difference of the noisy spectra incorporating the linear temperature gradient with respect to the reference spectrum is shown as well. Using the full interferogram, the mean difference shows only little differences across the spectral axis, which shows that the spectrum contains an averaged temperature information of the given temperatures within the field. Using only the left side of the interferogram
entails that the interferogram contains only temperatures from 190K to 200K. As explained in Sec. 4, lower temperatures means higher intensities in the central spectral region and lower intensities at the edges of the bandpass filter. Analogously, the same argument can be applied to the right side of the interferogram. The mean and standard deviation of the retrieved temperatures are shown in Figure 10a. The precision of the temperature retrieval is again decreased by $0.3K-0.4K$. Thus, the decrease in the temperature precision mainly comes from the mirrored shot noise. The retrieved temperatures of the single-
sided interferograms lie on average 6K apart of each other, which is closer than the mean temperature of each side suggests. The Fourier transformation maps a weighted sum of all spatial samples in the interferogram to a sample in the spectrum. Thus, the temperature information is localized in the interferogram, but fully distributed across the spectrum. The retrieved temperature is therefore an average of the temperature information within a given region of interest in the interferogram. Note that values deviating more from the mean in the interferogram contribute more to the spectrum and carry therefore more temperature
information into the spectrum. Thus, the large variations around the ZOPD in the interferogram contribute more to the overall result. Eq. (1) shows that the large variations comes from the fact that the interferogram consists of superimposed cosine waves with zero phase at ZOPD. This is the reason why we did not apply any apodization which applies higher weights to the central region of the interferogram. This will be discussed more in detail in Sec. 5.3.

## 5.1   Sensitivity to horizontal temperature variations

In this section we assess the sensitivity of the temperature retrieval to horizontal temperature variations. We define a function

$$f : \boldsymbol{T} \mapsto T_{ret} \tag{3}$$

which maps the horizontal temperature variation $\boldsymbol{T}$ to a retrieved average temperature $T_{ret}$. We calculate the derivative of $f$ with respect to $\boldsymbol{T}$ approximated by finite differences. Since the emission has different sensitivities at different background temperatures as shown in Fig. 4, the derivatives are calculated for the whole range of encountered temperatures. The result is
shown in Figure 11a. It shows an overall wavy pattern representing changing intensities of emission lines and its modulation through the instrument. The matrix shows that for any temperature level, the retrieved temperature is most sensitive to temperatures close to the main lobe. Figure 11b shows smoothed rows for selected temperature levels of the matrix in Fig 11a. Lower temperature below 300K show a lower sensitivity around the main lobe and more sensitivity to the side. Temperature levels around 500K have the highest central peak and the lowest sides. This effect is attenuated for temperatures above 500K. Thus,





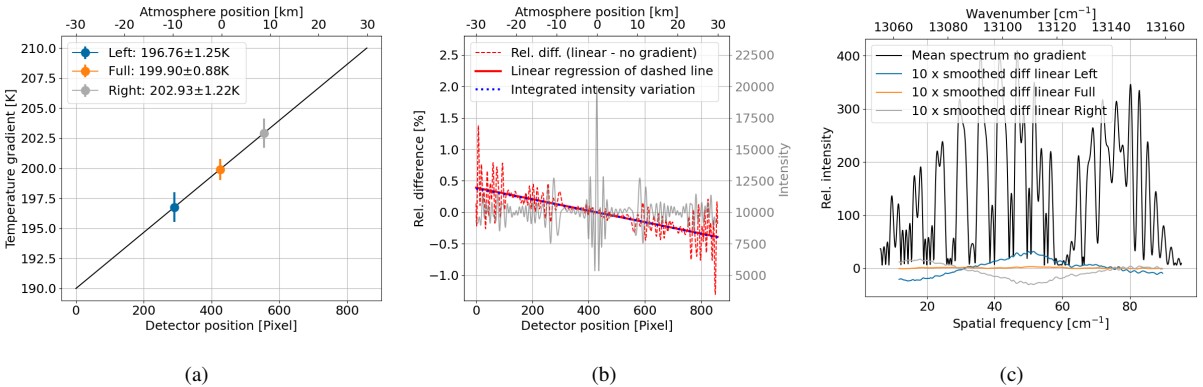

(a)           (b)           (c)

**Figure 10.** Results of Monte-Carlo simulations with 300 samples where shot noise is propagated through the spectrum into the temperature retrieval for an interferogram with a constant temperature at 200K and an interferogram with linear temperature gradient from 190K to 210K; (a) used temperature gradient and retrieved temperatures; (b) relative difference between interferogram without and with linear temperature gradient relative to the value 10 000 (mean of interferogram without temperature gradient); integrated intensity variation within the bandpass filter due to the temperature gradient relative to the intensity corresponding to the central temperature of 200K; (c) mean of noisy spectra without temperature gradient as reference; mean difference between the noisy spectra incorporating the linear temperature gradient and the reference mean spectrum using full and half-sided interferograms; no apodization is applied;

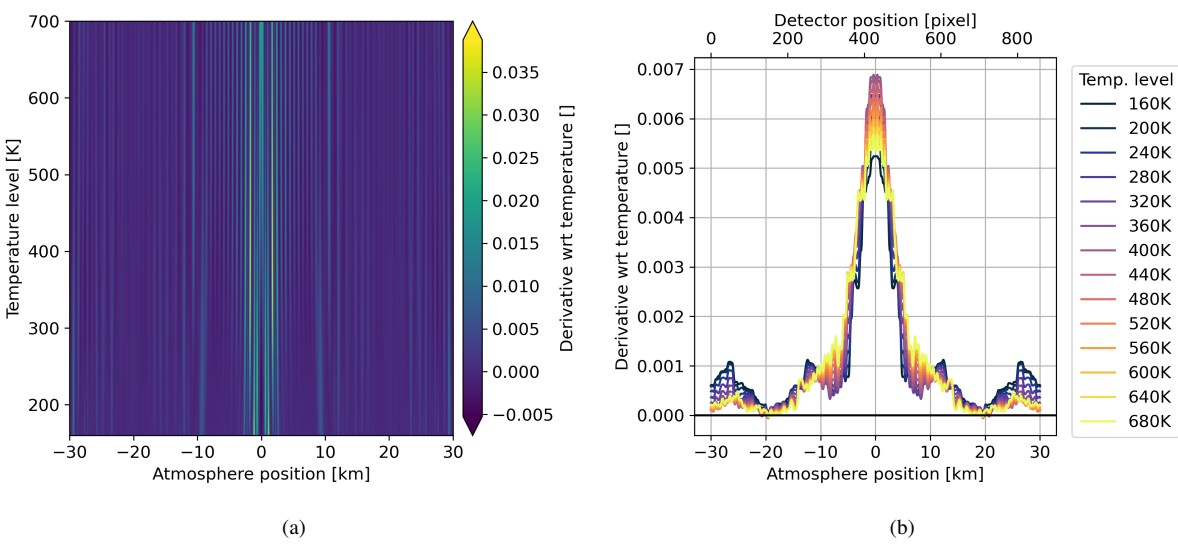

(a)                                  (b)

**Figure 11.** (a) Derivative of temperature at a given horizontal position for multiple temperature levels; (b) Selected rows of (a) and smoothed by a running mean with window size 101

the temperature retrieval is least sensitive to horizontal temperature variations around 500K. The same effect is seen in Sec. 5.2 and Sec. 5.3.





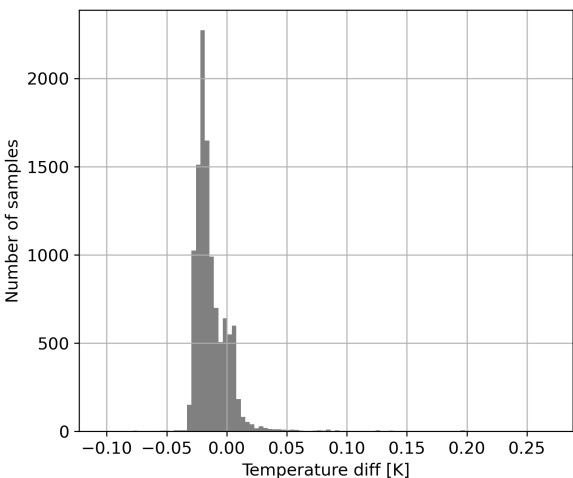

**Figure 12.** Temperature error using derivative matrix from Figure 11a and Eq. (4) to estimate retrieved temperature

The Jacobian matrix in Figure 11a can be used in Taylor's theorem to linearly approximate function $f$ defined in Eq. (3). We interpolate the 2-D field and derive the continuous derivative with respect to temperature variation. We split the temperature variation in background and residuals, denoted by $\boldsymbol{T} = \bar{T} + \boldsymbol{T}'$. The retrieved temperature can be then estimated due to Taylor's theorem by

$$\tilde{T}_{ret} = \left(\nabla f(\bar{T}) \cdot \boldsymbol{T}'\right) + \bar{T}, \tag{4}$$

where $\cdot$ denotes the scalar product of two vectors. To put it in context with the linearized diagnostic theory described by Rodgers (2000), the Jacobian of function $f$ is analogous to the averaging kernel matrix which also maps the atmospheric state to the retrieval result. The differences are that the function $f$ maps a vector to a scalar and thus the Jacobian of $f$ is a vector. Further, $f$ and therefore its Jacobian is dependent on the background temperature. The functionality of the presented estimation, however, is similar to the usage of the averaging kernel matrix in a usual retrieval.

To evaluate this approximation we use the simulated temperature variations from Sec. 5.2, estimate the retrieved temperature as explained and compare it to the retrieved temperature running through the entire simulation-processing process explained in Sec. 3.2 and Sec. 3.3. The results are shown in Figure 12, which shows an overall good agreement with $98\%$ being within an error of less than 0.05K. A slight but negligible bias of 0.02K can be seen. The minimal and maximal error is at $-0.10$K and 0.28K respectively. Thus, the explained method can be used to approximate the retrieved temperature for varying horizontal temperature variations without running through a full end-to-end simulation.

## 5.2 Temperature retrieval of horizontal temperature variations using single-sided interferograms

To get a comprehensive picture of the split interferogram processing method, we simulate interferograms according to temperature variations typically produced by gravity waves, split the interferogram and retrieve temperature for each side. When

**Figure 13.** Relative location of the retrieved temperatures within the temperature variation using single sided interferograms (a) for varying amplitude, (b) for varying temperature background level, (c) for varying horizontal wavelength and (d) for varying phase; the box extends from the lower to upper quartile values, the whiskers extend from the 5th to 95th percentile;





observing temperature variations produced by an atmospheric wave, it is essential to localize that information in space to obtain proper wave characteristics from that data. A sinusoidal horizontal temperature variation can be modelled by

$$\boldsymbol{T} = \bar{T} + A \cos\left(2\pi \frac{\boldsymbol{x}}{\lambda_h} + \phi\right) \tag{5}$$

where $\bar{T}$ is the background temperature, $A$ the amplitude, $\lambda_h$ the horizontal wavelength, $\phi$ the phase and $\boldsymbol{x}$ is the horizon-
tal scale. The horizontal field of view is assumed to be $\pm 30$km. Following Fig. 2a, we vary the temperature from 160K to 700K to cover typical low- to mid-latitude conditions. Following Chen et al. (2022), we alter the horizontal wavelength from 200km to 2000km and the amplitude from 4K to 30K. In total, we analysed 4220 simulated temperature variations. Note that temperature variations with a min-max value smaller than 1K are excluded, because they cannot be resolved. For the further analysis we introduce the term 'location' of a retrieved temperature. This is defined by the abscissa of that atmospheric model
temperature, which is equal to the retrieved one. The location of the retrieved temperatures of each side relative to the center are shown in Figure 13. Hereby, one parameter is varied while showing the distribution of all waves for the given parameter. Note that the distance between the locations of each side can be seen as a measure for how well the presence of a horizontal temperature variation can be characterized. Figure 13a shows no influence of the amplitude of the temperature residual onto the location of the retrieved temperatures. Figure 13b shows that the temperature background affects the distance of the retrieved
temperatures due to the different sensitivities of the $O_2$ A-band emission with respect to temperatures. The results in Sec. 5.1 explains the retrievals of lower temperature having a higher information content at the sides compared to that of higher temperatures and thus, resulting in retrieved temperatures laying further apart of each other when using single-sided interferograms. The minimum of this effect is around 500K with a slight increase for temperatures above. The horizontal wavelength of the temperature variation affects only slightly the distance of the retrieved temperature as shown in Figure 13c. Short wavelengths
results in a larger spread of the location distribution due to the fact that the phase of the temperature variation plays a greater role. When looking at the phase in Figure 13d, $\phi = 0$ and $\phi = \pi$ are outliers referring to the crest and trough of the captured temperature residual. In both cases, the temperature variation is low within the center and large at the edges. This works against the increased temperature information around the center and results in temperatures being further apart from each other. To conclude, if one takes the background temperature into account one can give a good estimate of the location of the retrieved
temperature. The effect of the phase will introduce a systematic error at the crest and trough. Compensating these two effects in the wave analysis, one gets a good estimate of the horizontal wave parameter components.

## 5.3  Apodization

In this section we asses the influence of apodization onto the retrieval of split interferograms. We evaluate this for horizontally linear temperature gradients with a spread of $\pm 10$K for central temperature levels from 160K to 700K and for the Norton-Beer
apodizations introduced in Sec. 3.3. The results are shown in Figure 14. Using the full interferogram the mean temperature can be recovered for each temperature level independent of the strength of the apodization. Using a single-sided interferogram, stronger apodization decreases the localization difference between the retrieved temperatures of the left and right side. Revis-





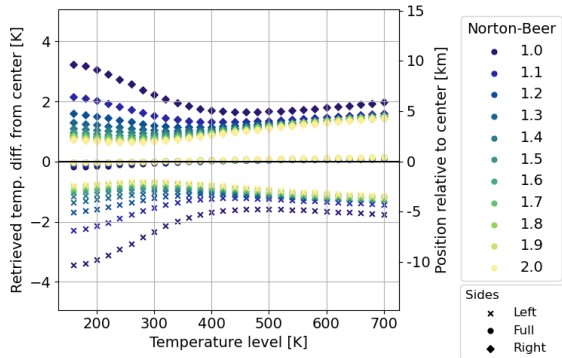

**Figure 14.** Retrieved temperatures using a linear temperature gradient for multiple temperature levels and different strengths of apodization

iting Figure 3a the apodization functions decreases the intensity of the interferogram towards the edges and thus put a greater weight on the information content in the central region. The effect that the retrieved temperatures are closer to the center as

explained in Sec. 5.1 is consequently amplified by the apodization function. The case without apodization (Norton-Beer 1.0) shows a decrease in the temperature difference between the two sides for temperature levels between 160K and 500K and a slight increase for higher temperatures. This shape is consistent with the results shown in Fig. 13b. When using mirrored single-sided interferograms, apodization is not only a trade off between spectral resolution and decrease of the side lobes, but also a trade off between spatial resolution of the two retrieved horizontal temperature data points and robustness against

errors (see Sec. 3.3 for link between apodization and robustness). Using single-sided interferograms therefore increases the requirements for instrument error mitigation, if the distance between the two retrieved temperature data points wants to be kept large.

## 6   Conclusions

Spatial heterodyne interferometers are often combined with a two-dimensional focal plane array and a telescope to obtain

spatial and spectral information from a scene. This study deals with a limb sounding SHI instrument, which delivers in its default configuration spatial information of the atmosphere in the vertical and spectral information in the horizontal direction across the LOS. However, it is possible to split the interferogram into half to obtain additional spatial information in horizontal direction across the LOS as well. This methodology is firstly applied to spatial heterodyne spectroscopy for atmospheric temperature, which then gives two horizontal temperature profiles (two temperature data points per tangent layer).

This paper first discussed the temperature sensitivity of the captured $O_2$ A-band emission and the resulting temperature precision of the instrument. A special focus is put on the upper tangent altitudes above 120km for day-time conditions. We simulated the expected signal levels for the given instrument specifications and present a temperature precision analysis for different temperatures and signal levels using full interferograms as a baseline simulation. It was shown that the temperature precision is negatively correlated with temperature due to the specific temperature dependence of the emission lines, so that



thermospheric temperatures are less precise than mesospheric ones. The simulations show that within the strong emission layer around 90km±10km the temperature precision stays below 1K. During day-time temperatures, altitudes up to 140km can be resolved with either a lower temperature precision or a lower spatial resolution. For example, one would need to bin 60 rows to resolve tangent altitudes around 140km with a temperature precision of 4K. Our analysis does not cover tangent altitudes below 80km, which need to be assessed by simulations using a radiative transfer model, which accounts for self-absorption.

Further, we show that the method of split interferograms decreases the temperature precision by a factor of $\sqrt{2}$. Analysing the influence of an horizontal temperature variation across the field of view shows that that the horizontal localization of retrieved temperatures is generally closer to the center of the field of view. A linearized estimation of the retrieved temperature for a given temperature variation using Taylor's theorem shows that most of the temperature information is localized around the center of the interferogram. Further, it is shown that apodization affects the spatial resolution of the data obtained by this

method. In general, weaker apodization gives better spatial resolution across the LOS, which must be balanced against model or instrumental uncertainties. As an application of this method, Chen et al. (2022) showed that medium-scale gravity waves can be horizontally resolved from such data, but it must be taken into account that the phase and background temperature of the captured wave affects the location of the retrieved temperatures.

**Appendix A: Spectral noise of a magnitudinal spectrum**

We can describe the noisy interferogram by

$$\hat{y}[n] = y[n] + \varepsilon[n] \tag{A1}$$

where $\hat{y}$ is the interferogram with noise, $y$ the non-noisy interferogram and $\varepsilon$ the shot noise described by the normal distribution

$$\varepsilon[n] \sim \mathcal{N}(0, \overline{y}) \tag{A2}$$

where $\overline{y}$ is the mean of the signal.

The shot noise can be propagated through the Fourier transformation, resulting in a noisy spectrum given by

$$\text{Re}(\hat{S}[k]) \sim \mathcal{N}(S[k], \frac{\overline{y}}{2N}) \tag{A3}$$

$$\text{Im}(\hat{S}[k]) \sim \mathcal{N}(0, \frac{\overline{y}}{2N}) \tag{A4}$$

where $N$ is the number of samples in the interferogram and $S$ is the non-noisy spectrum. The absolute value of the samples $\sqrt{\text{Re}(\hat{S}[k])^2 + \text{Im}(\hat{S}[k])^2}$ can be described by the Rice distribution (Talukdar and Lawing, 1991), where the distribution in

each spectral sample depends on the value of the non-noisy spectral sample. Let $\upsilon = S[k]$ be the mean distance of the real and imaginary part to the origin and let $\sigma = \sqrt{\frac{\overline{y}}{2N}}$ be the standard deviation of the real and imaginary part. The Rice distribution is defined by $\upsilon$ and $\sigma$ and the mean is given by



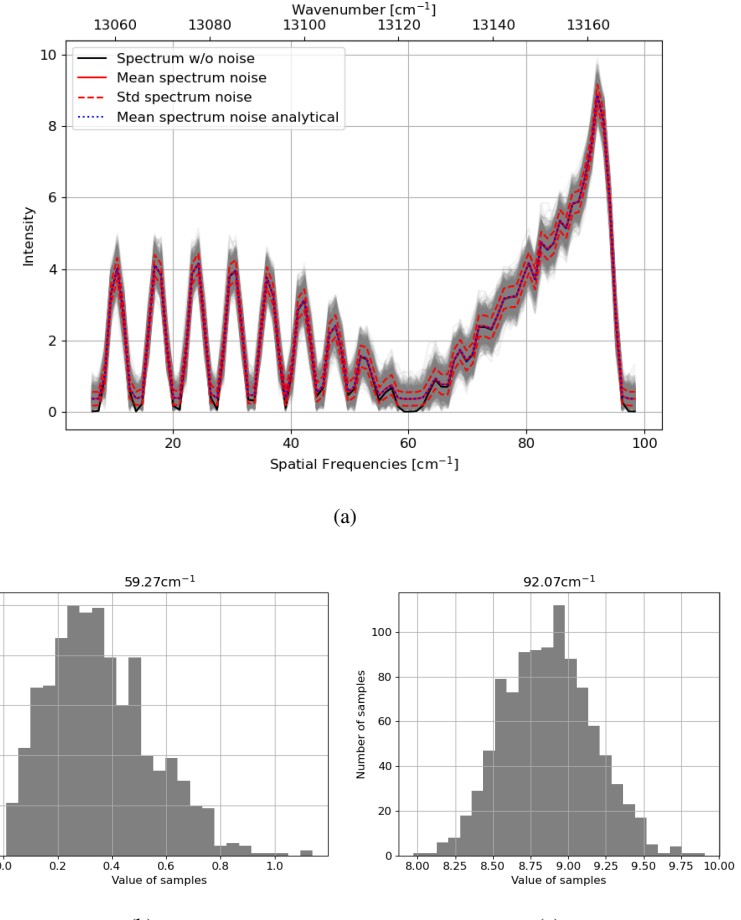

**Figure A1.** (a) Magnitudinal spectra for a low signal with an SNR of 20 and high temperature of 700K with (grey lines) and without (solid black line) shot noise; the mean and standard deviation of the noisy spectra (red solid and dashed lines) and the analytical mean calculation following the mean of the Rice distribution defined in Eq. (A5) (blue dotted line); (b) and (c) noise distribution at the minimal and maximal spectral sample at $59.27 \mathrm{cm}^{-1}$ and $92.07 \mathrm{cm}^{-1}$, respectively;





$$\mu = \sigma\sqrt{\frac{\pi}{2}}\,_1F_1\left(-\frac{1}{2};1;-\frac{v^2}{2\sigma^2}\right) \tag{A5}$$

where $_1F_1$ is the confluent hypergeometric function of the first kind. Note that if apodization is applied, the standard deviation
of the noise is decreased.

In Figure A1a we show magnitudinal spectra for a low signal with an SNR of 20 and high temperature of 700K. One can see that the noisy spectra are centered around the non-noisy spectrum for high values but are off for low values. Figure A1c shows that the noise distribution is close to a normal distribution for a high value but skewed for low values as shown in Figure A1b. Within the processing, the deviating mean is subtracted to at least center the noise distribution around the non-noisy spectrum,
which reduces the bias. Note that this affects only very low signals which are below the signal levels usually used within the processing.

*Author contributions.* KN performed all simulations and wrote most of the text. JU initiated the analysis regarding the sensitivity to horizontal temperature variations. JU and MK supervised the study. All authors contributed to the discussion of the results, the manuscript review and improvements.

*Competing interests.* At least one of the (co-)authors is a member of the editorial board of Atmospheric Measurement Techniques.

*Acknowledgements.* This project 19ENV07 MetEOC-4 has received funding from the EMPIR programme co-financed by the Participating States and from the European Union's Horizon 2020 research and innovation programme.



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
