# Peer review of "Observation of horizontal temperature variations by a spatial heterodyne interferometer using single-sided interferograms"

_Atmospheric Measurement Techniques, 2023_

## Author Comment (AC1)

**Response to the comments of referee #1**

We thank the referee for their valuable and helpful comments. We have addressed all of them one-by-one in details as listed below. The comments are in bold and our replies are in regular font. The line numbers indicated in our replies are given with respect to the track change manuscript, and may differ from the revised manuscript.

**General and specific points:**

**To split the 2-D interferogram into two single-side interferograms around the zero optical path difference is the key process to successfully derive two temperature profiles. The temperature information comes mostly from the center of the interferogram, so it would be expected that the temperature retrieval is sensitive to the determination of ZOPD. The authors may consider to add some retrieval results when ZOPD cannot be determined precisely, which could often happen during actual observations.**

The referee points out that it is crucial to know the location of the ZOPD. A sensitivity study was conducted by Ntokas et al. (2022), which showed that the ZOPD needs to be known on the sub-pixel scale, if it is not accounted for during the data processing or retrieval. This requirement is not meet in the raw data and therefore, correction methods needs to be applied. The feasibility of these methods is presented by Kleinert et al. (2014) and Ungermann et al. (2022). We added a discussion paragraph regarding this topic in Line 269-276.

**The text has tendency to omit the definite article "the" in some of its sentences. It is recommended that the authors review and add "the" where needed.**

The revised text includes the necessary definite articles where applicable.

**The colored lines in some figures are sometimes difficult to distinguish one from another, e.g., Fig.2(b), Fig. 10(b), Fig. A1(a), and**

**I would suggest to either change the colorbar or use markers if possible.**

We adjusted the above mentioned figures to increase their visibility. Specifically, for Fig.2(b) and Fig.10(b), we change the colors. Note for Fig.2 that on behalf of referee #2, the input temperature is shown separately for solar minimum and solar maximum conditions, and the production mechanisms and the estimated intensity count per pixel are shown individually for day- and night-time conditions. Furthermore in Fig.6, the temperature uncertainty is presented individually for day- and night-time simulations as well, to be consistent. For Fig.A1(a), we adjusted the colors and removed the standard deviation of the noisy spectra, which are not needed for the discussion. Discussions, captions and references of the figures are adjusted accordingly.

**Line 125-157: For O2 A band, self-absorption cannot be omitted below 90 km. When no self-absorption is assumed for above 80 km, it will affect the temperature retrieval to some extent between 80 km and 90 km. Authors may add some discussions on this.**

This comment led us to further investigation on this topic. In Fig. 1 we present the radiance distribution along the line of sight (LOS) normalized to the maximum of each LOS. The limit of 80km was previously derived by Fig. 1a. However, investigating the tangent altitudes between 80km and 90km reveals that the lower most altitudes are affected by self-absorption, where 50% from the radiance come from the strong signal region around 90km. We therefore agree with the referee and adjusted the manuscript in Line 137-144 and in the conclusion in Line 367-369.

[Figure]

Figure 1: Radiance distribution along the line of sight (LOS) normalized to the maximum of each LOS for (a) selected LOS over the full atmospheric vertical grid from 60km to 180km and (b) a zoom in of the tangent altitudes between 80km and 90km;

**Line 228-230: "values ..." is not clear in this sentence, please consider to rephrase/complement the sentence.**

We reformulated the sentence accordingly in Line 261-264.

**Technical comments and typos:**

- **Line 18: "...(Vincent (2015))..." would be "(Vincent, 2015)" and also in the other indirect citations in the text, e.g., L32.**

  We changed the text accordingly in Line 18, 44 and 107.

- **Line 20: "...summarize..." would be "summarized"**

  We changed the text accordingly in Line 20-21.

- **Line 22: "...point out..." would be "pointed out"**

  We added in the text accordingly in Line 22.

- **Line 22: "...underline...outlines... " would be "underlined...outlined"**

  We changed the text accordingly in Line 23-24.

- **Line 45: "if it it possible..." would be "if it is possible..."**

  We added in the text accordingly in Line 61.

- **Line 88: "...a electronic transition..." would be "an electronic transition..."**

  We changed the text accordingly in Line 104.

- **Line 89: "..., which" would be "...., and" /"..., where"**

  We changed the text accordingly in Line 105-106.

- **Line 132: "...which..." would be "whose"**

  We changed the text accordingly in Line 162.

- **Line 138: "...show..." would be "showed"**

  We changed the text accordingly in Line 169.

- **Line 139: "...1.4 and 1.6 refers..." to "1.4 and 1.6 refer"**

  We changed the text accordingly in Line 171.

- **Line 325: "...decreases..." to "reduces"**

  We changed the text accordingly in Line 370.

- **Line 326: duplicated "that" in the sentence**

  We changed the text accordingly in Line 371.

- **Line 333: "...affects..." to "affect"**

  We changed the text accordingly in Line 378.

**References**

A. Kleinert, F. Friedl-Vallon, T. Guggenmoser, M. Höpfner, T. Neubert, R. Ribalda, M. K. Sha, J. Ungermann, J. Blank, A. Ebersoldt, E. Kretschmer, T. Latzko, H. Oelhaf, F. Olschewski, and P. Preusse. Level 0 to 1 processing of the imaging Fourier transform spectrometer GLORIA: generation of radiometrically and spectrally calibrated spectra. *Atmospheric Measurement Techniques*, 7(12):4167–4184, Dec. 2014. ISSN 1867-8548. doi: 10.5194/amt-7-4167-2014. URL https://amt.copernicus.org/articles/7/4167/2014/.

K. Ntokas, M. Kaufmann, J. Ungermann, P. Preusse, and M. Riese. Retrieval of gravity wave parameters using half interferograms measured by CubeSats. In C. D. Norton and S. R. Babu, editors, *CubeSats and SmallSats for Remote Sensing VI*, page 9, San Diego, United States, Sept. 2022. SPIE. ISBN 978-1-5106-5456-3 978-1-5106-5457-0. doi: 10.1117/12.2633460. URL https://www.spiedigitallibrary.org/conference-proceedings-of-spie/12236/2633460

J. Ungermann, A. Kleinert, G. Maucher, I. Bartolomé, F. Friedl-Vallon, S. Johansson, L. Krasauskas, and T. Neubert. Quantification and mitigation of the instrument effects and uncertainties of the airborne limb imaging FTIR GLORIA. *Atmospheric Measurement Techniques*, 15(8): 2503–2530, Apr. 2022. ISSN 1867-8548. doi: 10.5194/amt-15-2503-2022. URL https://amt.copernicus.org/articles/15/2503/2022/.

---

## Author Comment (AC2)

**Response to the comments of referee #2**

We thank the referee for their review including detailed comments and suggestions. It will strengthen the output of the study. We have addressed all of them one-by-one in details as listed below. The comments are in bold and our replies are in regular font. The line numbers indicated in our replies are given with respect to the track change manuscript, and may differ from the revised manuscript.

**General issues:**

**You absolutely cannot ignore self-absorption in the 80-85 km region. Even in the 85-90 km region it is not negligible. If you want to include this region (80-90 km), you must account for self absorption.**

This comment led us to further investigation on this topic. In Fig. 1 we present the radiance distribution along the line of sight (LOS) normalized to the maximum of each LOS. The limit of 80km was previously derived by Fig. 1a. However, investigating the tangent altitudes between 80km and 90km reveals that the lower most altitudes are affected by self-absorption, where 50% from the radiance come from the strong signal region around 90km. We therefore agree with the referee and adjusted the manuscript in Line 137-144 and in the conclusion in Line 367-369.

[Figure]

Figure 1: Radiance distribution along the line of sight (LOS) normalized to the maximum of each LOS for (a) selected LOS over the full atmospheric vertical grid from 60km to 180km and (b) a zoom in of the tangent altitudes between 80km and 90km;

**It is not enough to simply say you're "using HITRAN" to forward model the line intensities. At line 147, you say that you convolve the line strengths with the ILS. You've skipped a few steps here. How are you accounting for broadening? What types of broadening are you accounting for? Are you actually just convolving the line strengths? Because you need to convolve the emission spectrum (which you calculate from the line strengths), see Babcock and Herzberg, 1948 (doi:10.1086/145062).**

The forward model has been tested for Doppler broadening referring to a Gauss shape and Doppler and pressure broadening referring to a Voigt shape. The results are depicted in Figure 2. The spectrally integrated radiance is examined in Figure 2a. It is observed that the simulation using the Gaussian line shape exhibits slight deviations for tangent altitudes below 80km. Nevertheless, these deviations are extremely small and can be neglected. The slightly enhanced flanks of the Voigt line shape, attributed to the pressure-induced Lorentzian shape, become apparent only when the differences are amplified, as demonstrated in Figure 2b. Thus, only Doppler broadening is considered in the forward model.

[Figure]

(a)              (b)

Figure 2: (a) Spectrally integrated radiance using Voigt and Gaussian line shape; right panel shows the difference of the simulation using a Gaussian line relative to the simulation using Voigt line shape; (b) strongest emission line for tangent altitude 60km using the Voigt line shape compared to the same emission line in the simulation using Gaussian line shape; the difference (Voigt - Gaussian) is amplified by a factor of 100;

Regarding the convolution of the atmospheric spectrum with the instrument line shape (ILS), it should be noted that the emission lines are extremely narrow compared to the ILS width as shown in Figure 3, and thus can be approximated by a Dirac impulse. The convolution of a function with a Dirac impulse is the function itself and thus, the ILS can be positioned at the position of the emission line and scaled by the line strength. Figure 3 shows that the two methods show only small differences and retrieve the same temperature, where the line strength method is used in the forward model for both cases. Furthermore in this study, the interferogram is built from the line strength it self, as shown in Eq.(1). Thus the forward calculation and the retrieval is consistent in itself. Some discussion is added in Line 157-162.

[Figure]

Figure 3: Normalized atmospheric spectrum with resolved narrow emission lines of a homogeneous gas cell for temperature equal to 200K; 'Convolution' refers to the atmospheric spectrum convoluted with the ILS; 'Line strength' refers to the method presented in the paper, where the ILS of each emission line is scaled with the line strength; Temperature indicated in the labels are the retrieved temperature using line strength method in both cases;

The discussion of temperature precision is good. However, a discussion on accuracy is also needed, especially for the daytime retrievals. Specifically, on how you're going to deal with background solar radiation and stray light, and how those will affect the accuracy of the temperature retrievals. It's only at altitudes very close to 90 km where the background solar signal is somewhat negligible compared to the airglow signal. And, if this is intended to be on a nanosat, you're likely going to have limitations on the size of baffle you can use, which means stray light will certainly be an issue. The source of that stray light will be from the bright Earth below, which will have a complicated self-absorption A-band signal, ie, it's not a simple linear function across the spectrum that you need to subtract. These background signals need to be accounted for and discussed.

We agree with the referee that the day-time observations are affected by direct solar radiation and stray light. First sensitivity studies were conducted recently, which showed that the baffle is long enough to neglect direct solar radiation if the sun is not in or very close to the field of view. Stray light due to upwelling radiation specifically from the ground however, affects largely the lower and upper tangent altitudes. Further investigations on this topic and possible correction methods need to be developed for an accurate temperature estimation. This however will not be included in this study. This study mainly focuses on the retrieval of horizontal temperature variations. A small discussion on this is added in Line 150-153.

**Specific issues:**

**Introduction: there have been two instruments launched recently that also use the A-band to measure MLT temperatures, MIGHTI on ICON, and the Swedish MATS satellite instrument. Please mention/reference these as well.**

We considered the referee's suggestion and added some information of MIGHTI and MATS instrument in Line 30-37.

**Line 26: This sentence is quite vague, please elaborate on why/where/how the instrument was developed.**

We elaborated more on the development process of the instrument in Line 37-41.

**Lines 26-28: This section is somewhat misleading. It sounds like you're saying that the first instrument (described in Kauffman et al. 2018) was successful in measuring temperature profiles. In that paper, it says that the instrument worked nominally on a rocket launch, however, wasn't able to produce temperature profiles. And the second part of this section makes it sound like a second instrument has been built and is ready to be tested. Is this the case? It should be made clear that Chen et al. 2022 is a simulation study.**

We restructured the section in Line 37-41 and Line 45-49 to address this comment in accordance with the previous comment. Furthermore, it is made clear that Chen et al. 2022 is a simulation study in Line 60-61.

**Line 55 (and throughout text): "asses" should be "assess"**

We corrected the spelling in Line 70, 71, 202 and 336.

**Figure 2: It would be helpful to split these into solar max and solar min in different plots. Also, maybe separate daytime and nighttime**

We welcome the suggestion of the referee and split the presented 1-D temperature profile into solar minimum and solar maximum condition. Furthermore, the production mechanisms and the expected intensity count per pixel are split into day- and night-time simulations, respectively in Fig.2b,e and 2c,f. Also, the colors has been changed of Fig.2b to address referee #1. Furthermore in Fig.6, the temperature uncertainty is presented individually for day-time and night-time simulations as well, to be consistent. Discussions, captions and references of the figures are adjusted accordingly.

**Fig. 10b: the legend should also include the grey interferogram with no gradient**

Fig. 10b and its associated caption has been updated, to increase its comprehensibility.

**Line 316: I don't recall any special attention being given to results above 120 km. Is this the intended altitude?**

120km was the upper limit of the vertical field of view of previous publications (Chen et al., 2022; Kaufmann et al., 2018). This is the first simulation study, which explores the upper limit mainly during day-time conditions, as the lower part of the field of view is affected by self-absorption and stray light from the ground. However, we agree with the referee that the formulation can be misleading. We therefore reformulated the sentence in Line 359-360.

**References**

Q. Chen, K. Ntokas, B. Linder, L. Krasauskas, M. Ern, P. Preusse, J. Ungermann, E. Becker, M. Kaufmann, and M. Riese. Satellite observations of gravity wave momentum flux in the mesosphere and lower thermosphere (MLT): feasibility and requirements. *Atmospheric Measurement Techniques*, 15(23):7071–7103, Dec. 2022. ISSN 1867-8548. doi: 10.5194/amt-15-7071-2022. URL https://amt.copernicus.org/articles/15/7071/2022/.

M. Kaufmann, F. Olschewski, K. Mantel, B. Solheim, G. Shepherd, M. Deiml, J. Liu, R. Song, Q. Chen, O. Wroblowski, D. Wei, Y. Zhu, F. Wagner, F. Loosen, D. Froehlich, T. Neubert, H. Rongen, P. Knieling, P. Toumpas, J. Shan, G. Tang, R. Koppmann, and M. Riese. A highly miniaturized satellite payload based on a spatial heterodyne spectrometer for atmospheric temperature measurements in the mesosphere and lower thermosphere. *Atmospheric Measurement Techniques*, 11(7):3861–3870, July 2018. ISSN 1867-8548. doi: 10.5194/amt-11-3861-2018. URL `https://amt.copernicus.org/articles/11/3861/2018/`.

---

## Author Response (AR2)

**Response to the comments of referee #2**

We thank the referee for their helpful comments. With the revised version, we tried to incorporate the reviewer's comments into the revised manuscript. We have addressed all of them one by one in detail as listed below. The reviewer's comments are in bold, and our replies are in regular font. The line numbers indicated in our replies are given with respect to the track change manuscript and may differ from the revised manuscript.

**General issues:**

**Lines 147-149: HITRAN does not give you the emission spectra. HITRAN only gives you the line strengths and parameters at 296 K. You then need to scale those to the line strengths for different temperatures (the equation for which, unless I'm mistaken, is not in the Gordon et al paper you reference). But more importantly, you then need to convert the temperature dependent line strength values into an actual emission energy spectrum. If you are only using HITRAN line strengths, you have not calculated proper emission spectra, see Babcock and Herzberg (1948) doi:10.1086/145062, and references therein (esp. Herzberg, 1939).**

We agree with the referee that the previous text lacked clarity in conveying the intended information. As a response, we have made revisions to the relevant section. Furthermore, we have incorporated a reference to an equation that outlines the calculation for relative line strength. This calculation utilizes spectroscopic parameters of each emission line, which are taken from the HITRAN data set. The revised version can be found in line 161–163, which now reads as follows: 'Instead of calculating the full radiative transfer equation, it calculates the relative distribution of the oxygen A-band emission lines for a given temperature following Song et al. (2017), where the required spectroscopic parameters of the emission lines are taken from the HITRAN data set.'

**Minor issues:**

**Line 26: missing "et al." in the reference**

We would like to point out, that the reference is a PhD thesis and therefore is only written by one author.

**Lines 33-40: This section still is confusing, and I think it's because the instrument doesn't have a name. I think the solution is to describe the instrument in the first sentence. Please be very specific as to what kind of limb instrument was built and for what purpose in the first sentence. It would also help to make this a separate paragraph from the preceding sentences.**

This comment is addressed in accordance with the subsequent comment. The changes are described below the next comment.

**Line 46: What do you mean by "Our instrument"? Have you built an instrument? Please be clearer before talking about "your instrument" whether it's one you've built, or one you've designed and planning to build, or whether you're simply talking about the same instrument from Chen et al. If this is in fact your instrument, I'd highly recommend giving it a name in order to clear up some of this confusion.**

To address the latter two comments, we have restructured the entire section, retaining the same content as previously presented. However, we have relocated certain parts to enhance comprehension. Additionally, we have introduced paragraphs to enhance readability. Furthermore, we have included the instrument's name to provide readers with a reference point. You can review the revised section in line 35–64.

**Line 55: What kind of resolution?**

The references used single sided interferogram to enhance spectral resolution. The revised version has been changed accordingly by adding the word 'spectral' in line 68.

**Figure 1 caption: "SHS" should be "SHI"**

We thank the reviewer to spot this error. The caption of Figure 1 has been changed accordingly in the revised version.

**Line 101-102: This sentence makes it sound like only HITRAN can be used to calculate the absorption and emission spectra. I'd suggest making it two sentences, and have the second start with something like, "In order to calculate the emission energies between rotational states, . . . "**

We welcome the referees suggestion and changed the text in the revised version accordingly in line 115–116, which now reads as follows: 'The band consists of multiple emission lines due to the transition of multiple rotational states. In order to calculate the emission energies between the rotational states, the HITRAN database can be used (Gordon et al., 2022).'

**Line 103: There are many different excited O2 states. Please be specific and replace "excited $O_2$" with "$O_2(^1\Sigma)$" (or another common notation) here and throughout the manuscript.**

The text in the revised version has been changed accordingly in line 116 and line 118.

**Line 109: A-band "emissions" should probably be "volume emission rates" as to not be confused with emission energies.**

The referee is correct and the text in the revised version has been changed accordingly in line 123.

**Line 136: Please be a bit more specific with altitude range, I'd highly recommend, ". . . the lowermost tangent altitudes, below 85 km, need to be treated with reservation."**

The text in the revised version has been changed accordingly in line 150.

**Line 149: Why are the spectra scaled and what are they scaled to?**

In the Forward model, the spectrum is scaled with a scaling factor to match the output spectrum from the Fourier transform. In this context, the factor is a retrieval parameter and corresponds to the number density of excited $O_2(^1\Sigma)$ molecules. This information has been added to the revised manuscript in line 164–165, which now reads as follows: 'Subsequently, it convolves the emissions with a given instrument line shape (ILS) and scales the total spectrum with a scaling factor to match the output spectrum from the Fourier transform. In this context, the factor corresponds to the number density of excited $O_2(^1\Sigma)$ molecules.'.

**References**

I. Gordon, L. Rothman, R. Hargreaves, R. Hashemi, E. Karlovets, F. Skinner, E. Conway, C. Hill, R. Kochanov, Y. Tan, P. Wcisło, A. Finenko, K. Nelson, P. Bernath, M. Birk, V. Boudon, A. Campargue, K. Chance, A. Coustenis, B. Drouin, J. Flaud, R. Gamache, J. Hodges, D. Jacquemart, E. Mlawer, A. Nikitin, V. Perevalov, M. Rotger, J. Tennyson, G. Toon, H. Tran, V. Tyuterev, E. Adkins, A. Baker, A. Barbe, E. Canè, A. Császár, A. Dudaryonok, O. Egorov, A. Fleisher, H. Fleurbaey, A. Foltynowicz, T. Furtenbacher, J. Harrison, J. Hartmann, V. Horneman, X. Huang, T. Karman, J. Karns, S. Kassi, I. Kleiner, V. Kofman, F. Kwabia–Tchana, N. Lavrentieva, T. Lee, D. Long, A. Lukashevskaya, O. Lyulin, V. Makhnev, W. Matt, S. Massie, M. Melosso, S. Mikhailenko, D. Mondelain, H. Müller, O. Naumenko, A. Perrin, O. Polyansky, E. Raddaoui, P. Raston, Z. Reed, M. Rey, C. Richard, R. Tóbiás, I. Sadiek, D. Schwenke, E. Starikova, K. Sung, F. Tamassia, S. Tashkun, J. Vander Auwera, I. Vasilenko, A. Vigasin, G. Villanueva, B. Vispoel, G. Wagner, A. Yachmenev, and S. Yurchenko. The HITRAN2020 molecular spectroscopic database. *Journal of Quantitative Spectroscopy and Radiative Transfer*, 277: 107949, Jan. 2022. ISSN 00224073. doi: 10.1016/j.jqsrt.2021.107949. URL https://linkinghub.elsevier.com/retrieve/pii/S0022407321004416.

R. Song, M. Kaufmann, J. Ungermann, M. Ern, G. Liu, and M. Riese. Tomographic reconstruction of atmospheric gravity wave parameters from airglow observations. *Atmospheric Measurement Techniques*, 10(12):4601–4612, Nov. 2017. ISSN 1867-8548. doi: 10.5194/amt-10-4601-2017. URL https://amt.copernicus.org/articles/10/4601/2017/.